# Alcohol and Tobacco Consumption, Personality, and Cybervictimization among Adolescents

**DOI:** 10.3390/ijerph16173123

**Published:** 2019-08-28

**Authors:** Mónica Rodríguez-Enríquez, Miquel Bennasar-Veny, Alfonso Leiva, Aina M. Yañez

**Affiliations:** 1Child and Adolescent Psychiatry and Psychology Department, Hospital Sant Joan de Déu of Barcelona, Paseig de Sant Joan de Déu, 2, 08950 Barcelona, Spain; 2Nursing and Physiotherapy Department, University of the Balearic Islands, Cra. de Valldemossa, Km 7,5, 07122 Palma, Spain; 3Primary Care Research Unit of Mallorca, Balearic Islands Health Service, C/Escuela Graduada, 3, 07002 Palma, Spain; 4Research Group on Global Health & Human Development, Balearic Islands University, Cra. de Valldemossa, Km 7,5, 07122 Palma, Spain

**Keywords:** cyberbullying, cybervictimization, personality, adolescents, alcohol, tobacco, school

## Abstract

Cyberbullying has emerged as a public health problem. Personality may play an important role in substance use and cybervictimization. The aim of this study was to examine whether tobacco and alcohol consumption and personality traits are associated with cybervictimization in Spanish adolescents. A cross-sectional study was conducted with 765 secondary students (aged 14–16) from 16 secondary schools in Spain. Participants completed a questionnaire assessing sociodemographic characteristics; tobacco and alcohol consumption; cybervictimization (Garaigordobil Scale); and personality traits (Big Five Questionnaire). A logistic regression model controlling for sex, age, parental education and personality traits was used to determine the independent associations and interactions between tobacco and alcohol consumption and cybervictimization. The results indicate that a total of 305 adolescents (39.9%) reported that they were cyberbullied in the past year. Girls were more likely to be cyberbullied than boys. Cybervictims had a significantly greater monthly alcohol consumption (OR = 1.51; 95% CI = 1.05–2.15), higher scores for extraversion (OR = 1.31; 95% CI = 1.06–1.63) and emotional instability (OR = 1.53; 95% CI = 1.27–1.83); as well as lower scores for conscientiousness (OR = 0.78; 95% CI = 0.63–0.95). These results suggest that personality traits and alcohol consumption are independently associated with cybervictimization. Our study suggests the existence of underlying common personality factors for cybervictimization and alcohol and tobacco use.

## 1. Introduction

Cyberbullying and substance use have a powerful negative effect on young people’s health and well-being [1]. Cyberbullying is an online (or using electronic forms of contact), intentional aggression by one or more individuals that repeatedly targets people who cannot easily defend themselves [2]. Victims of cyberbullying are called cybervictims. A significant proportion of children and adolescent (20–40%) are cybervictims [3,4]. Previous studies have found sex differences in cyberbullying and cybervictimization—girls were more likely than boys to experience cyberbullying [5]. However, the existing evidence showed mixed results and the majority of studies did not find any sex differences [6]. Cybervictimization and cybervictims have emerged as a new public mental health problem among adolescents in the last two decades and it increases the risk of suicide-related behaviors [7].

A recent review identified conduct problems and social problems as the main predictors of victimization in adolescence and concluded that more research is required for clarification [8]. Cross-sectional studies have shown a two- to three-fold higher odds of being cybervictimized among alcohol and tobacco users [9,10,11]. Longitudinal studies show that substance use, including alcohol and tobacco, predict cybervictimization and not vice versa [12,13].

Personality traits play an important role as a risk factor for substance use and cybervictimization. Higher levels of neuroticism (e.g., feelings of anxiety; fear; worry and depression; having low self-esteem; and use of maladaptive and/or impulsive strategies to cope with stress) and low levels of conscientiousness (e.g., tending to behave in careless, irresponsible and lazy ways) are related to a higher risk of substance abuse [14]. A lower level of childhood conscientiousness predicts smoking in adulthood; and a higher level of childhood extraversion (e.g., tendency to be sociable, energetic, talkative) and neuroticism predicts alcohol use in adulthood [15]. Higher neuroticism, openness to experience (e.g., curiosity, need to experience new things) and extraversion levels predict alcohol dependence and higher neuroticism and openness to experience; while lower responsibility levels predict tobacco dependence among college students. However, the personality traits associated with substance abuse are different for girls and boys [16].

A meta-analytic review of bullying victimization (without discrimination between different types of bullying) has found an association with a lower level of agreeableness and conscientiousness, and higher levels of neuroticism and extraversion [17]. However, there are few studies about cybervictimization and Big Five personality traits, and the evidence is not consistent. Cybervictimization has been positively associated with a higher level of neuroticism and openness and a lower level of conscientiousness and agreeableness [18]. Other studies found that only extraversion and openness were significant predictors of cyberbullying victimization [19]. The personality traits related to cybervictimization may also be different for boys and girls [17].

These findings are consistent with the Problem Behaviour Theory [20]. The conceptual structure of this theoretical framework includes different domains (social environment, perceived environment, personality, behavior and biology) that are intercorrelated and determine clusters of risk behaviors that could be explained by similar antecedents [21]. In line with this theory, adolescents who engage in antinormative behaviour, such as substance use, could easily break other rules (e.g., skipping class, getting into fights, shoplifting) and spend more time with antisocial peers with a higher risk of getting involved in other problems such as cybervictimization.

Cybervictimization, substance use, and their relationship could be explained by common underlying personality traits. The purposes of the present study are to examine the association of cybervictimization with personality, with tobacco and alcohol use, and to evaluate potential sex-mediated differences. According to the hypothesis of the Problem Behaviour Theory, we expected to find an association between substance use and cybervictimization. Moreover, we hypothesized that neuroticism (emotional instability), extraversion, and conscientiousness would make both a direct and an indirect—through substance use—contribution to cybervictimization.

## 2. Materials and Methods

### 2.1. Sample and Procedure

The study participants were students aged 14–16 years who participated in the ITACA (“Intervenció multifactorial orientada a disminuir la prevalença de Tabaquisme a la població Adolescent: Assaig Clínic Aleatorizat per clusters”) project: A multi-center, cluster-randomized, controlled trial aimed at reducing the prevalence of smoking among secondary education students in Spain [22]. The initial ITACA sample comprised 1708 students (11–12 years-old) from 16 secondary education schools covering a wide range of communities (urban, semi-urban and rural), socioeconomic status and prevalence of smoking. The schools were randomly assigned to a 4-year curriculum-based multifactorial intervention or control groups. In this study, we focused on the third wave of assessment (September–December 2015) when personality and cybervictimization were assessed (1230 students). Participants met the inclusion criteria if they attended school on the day of the survey, if their parents agreed with their participation in the study, and if the student could be identified and matched with baseline data. The final sample comprised of 765 students.

Students completed surveys during a 45-min class in grade 4 of their secondary education. The surveys were administered by two trained data collectors. The teachers were asked to leave the classroom during the surveys to ensure that the responses were confidential. Written informed consent was obtained from all students and from at least one parent/guardian prior to administering the survey.

### 2.2. Measures

#### 2.2.1. Sociodemographic Characteristics

Age, sex and the education level of both parents were recorded. The four categories of parental education were (a) less than 6 years of primary education, (b) 6 to 8 years of primary education, (c) 4 to 6 years of secondary education, and (d) university degree. For analysis of these data, a dichotomous variable was used (primary or less/secondary or more).

#### 2.2.2. Drug Use

Smoking status was assessed using seven items adapted from a previously validated questionnaire designed to assess smoking behaviors in adolescents [18]. Information on tobacco use was collected through the following question: “Which of the following statements best describes you? (1) I have never tried to smoke; (2) I have tried cigarettes a few times, but I do not smoke now; (3) I currently smoke at least one cigarette per month, but less than one cigarette per week; (4) I currently smoke at least one cigarette per week; (5) I smoke every day; (6) I used to smoke regularly in the past, but I do not smoke now”. The smoking statuses of adolescents were classified into non-smokers (those who answered 1, 2 or 6) and monthly smoking or more (3, 4 or 5). Alcohol consumption was assessed using the following question: “How often do you drink alcohol (including beer, wine, shots, brandy, rum, gin, whisky, etc., and mixes with soft drinks? (1) Never-hardly ever; (2) A few times a month; (3) Every weekend; (4) Every day. Alcohol consumption was classified into non-drinkers (1) and monthly drinking or more (2, 3 or 4)”.

#### 2.2.3. Personality Traits

Personality was assessed using the Big Five Questionnaire for Children (BFQ-C) [23], which has 65 questions that assess five basic personality traits: extraversion, agreeableness, conscientiousness, openness, and emotional instability (neuroticism). Extraversion questions assess characteristics such as activity, enthusiasm, assertiveness, and self-confidence. Agreeableness questions assess concern and sensitivity towards others and their needs. Conscientiousness questions assess dependability, orderliness, precision, and the fulfilment of commitments. Emotional instability questions assess feelings of anxiety, depression, discontentment, and anger. Openness questions assess self-reported intellect, especially in the school domain, and broadness or narrowness of cultural interests and fantasy/creativity. Previous studies have shown the good psychometric properties of this questionnaire [24].

#### 2.2.4. Cybervictimization Assessment

Cybervictimization was evaluated using the Garaigordobil Cybervictimization Scale [18], an instrument validated in the Spanish population. Previous psychometric studies have confirmed its validity and reliability, and it has a high Cronbach alpha (α = 0.82). Internal consistency in the sample of the present study was adequate (α = 0.83). Cybervictimization was assessed by asking students about the frequency (0 = never, 1 = sometimes, 2 = often, 3 = always) of suffering 15 cyberbullying behaviors during the last 12 months. The specific behaviors can be summarized as: Sending offensive and insulting messages; making offensive phone calls; recording an assault and uploading it to the internet; spreading photos or videos of embarrassing situations; taking stolen photos and spreading them online; making anonymous frightening phone calls; blackmailing by phone or internet; sexually harassing someone; spreading rumors, secrets, or lies; stealing a password; sending altered photos or videos to the internet; harassing or isolating someone from a social network; blackmailing in order to not divulge intimate details; threatening to kill others; defaming or telling lies to discredit others.

Cybervictimization was classified into two categories: (1) Adolescents who declared that they had suffered some type of harassment at least “sometimes” during the previous year; and (2) adolescents who never suffered harassment.

### 2.3. Statistical Analysis

To test whether personality traits, drug use and sociodemographic characteristics differ across cybervictimization categories, student *t*-test and Chi squared tests were performed. Four logistic regression models were fitted to determine the independent associations of personality traits (extraversion, agreeableness, conscientiousness, emotional instability and openness) and drug use with cybervictimization. Odds ratios (OR) with 95% confidence interval (CI) were calculated for personality z-scores (SD = 1). The dependent variable was cybervictimization, categorized as at least one time or never. All models were controlled for sex, age and the education level of both parents. All possible statistical interactions of independent variables (drug use and each personality trait) with sex were tested, fitting additional logistic models with interaction terms. The adequacy of regression models was based on Hosmer–Lemeshow’s goodness-of-fit and the area under the Receiver Operating Characteristics (ROC) curve. All analyses were performed on Statistical Package for Social Science (SPSS) version 22.0 (IBM Company, New York, NY, USA) for Windows.

### 2.4. Ethical Aspects

The research protocol was approved by the Primary Care Research Committee and the Institutional Review Board of the Balearic Islands Health Service (CEI-IB Ref. No: 1146/09PI). The study was conducted according to the ethical guidelines of the Declaration of Helsinki. Written informed consent was obtained from all students and at least one parent per student. All materials and procedures were approved by the educational authority.

## 3. Results

Descriptive statistics are shown in Table 1. Of the 765 adolescents included in the study, 305 (39.9%) reported that they had been cyberbullied in the past year. There were no significant differences in age (cybervictims 14.95 ± 0.08 vs. non-cybervictims 15.03 ± 0.06; *p* = 0.110) or parents’ educational levels (mother’s *p* = 0.874; father’s *p* = 0.538). Girls were more likely to be cyberbullied than boys (43.1% vs. 35.7%; *p* < 0.05).

The cybervictims had significantly greater alcohol (61.4% vs. 43.2%; *p* < 0.001) and tobacco consumption (7.9% vs. 3.5%; *p* < 0.01). The cybervictims had higher scores for extraversion (0.11 ± 1.03 vs. −0.094 ± 0.95; *p* < 0.01) and emotional instability (0.162 ± 1.02 vs. −0.234 ± 0.90; *p* < 0.001), and lower scores for conscientiousness (−0.001 ± 1.00 vs. 0.200 ± 0.96; *p* < 0.01).

To examine the independent associations of adolescent personality and alcohol and tobacco consumption with cybervictimization, we fitted a set of four-stage logistic regression models (Table 2). Adjusting for the effects of age, sex and parental education; higher extraversion (OR = 1.45; 95% CI = 1.19–1.78), emotional instability (OR = 1.58; 95% CI = 1.32–1.89) and lower conscientiousness (OR = 0.73; 95% CI = 0.60–0.89) were significantly associated with higher risk of cybervictimization (Model 1). Furthermore, alcohol consumption (OR = 1.99; 95% CI = 1.47–2.70) and tobacco consumption (OR = 2.55; 95% CI = 1.28–5.05) were also associated with cybervictimization (Model 2 and Model 3, respectively). When personality and substance use were taken together, the associations for conscientiousness and emotional instability remained significant and similar in magnitude to those observed before (Model 4), while the association for extraversion was somewhat lower (OR = 1.31; 95% CI = 1.06–1.63), that for alcohol use was lower (OR = 1.51; 95% CI = 1.05–2.15) and that for tobacco consumption was not significant anymore (OR = 1.51; 95% CI = 0.73–3.14).

To test whether sex moderated the associations between students’ personality traits, alcohol consumption, and cybervictimization we fit additional logistic models with all possible sex interaction terms. In these models, none of the interactions between personality and parental education were statistically significant (Table 2).

## 4. Discussion

Our results indicate that personality traits and alcohol consumption were independently associated with being cybervictimized. Students who had high scores for neuroticism or extraversion, or low scores for conscientiousness also had a greater risk for cybervictimization.

In consonance with Problem Behaviour Theory [20], which points out how adolescent risk behaviors do not occur randomly but tend to cluster and share common psychosocial risk factors, adolescents with higher levels of neuroticism or extraversion or low levels of conscientiousness could use coping strategies that increase the risky behaviors associated with both substance use and cybervictimization. In this sense, personality could be a common cause for being cybervictimized and for the consumption of alcohol and tobacco. Moreover, after removing the effect of personality by a multivariate analysis, we observed that alcohol consumption also remained independently associated from being victimized.

Furthermore, the association between extraversion and being cybervictimized was partially mediated by alcohol consumption. In the multivariate model, after adjustment by alcohol, there was a decrease in the magnitude of the association between higher levels of extraversion and being cybervictimized. Extraverted people have a preference for seeking, engaging in, and enjoying social interactions, and they share their emotions with others as an emotional coping strategy. Social interactions and sharing emotions have been associated with positive mental health and proper emotional adjustment [25], which could be a protective factor against traditional bullying victimization [26]. However, extraverted individuals experience a higher mood-enhancement from alcohol than non-extraverted people when they are with other people [27]. Thus, the greater preference to openly share their thoughts and emotions could be non-adaptive when personal information is being shared with non-trustworthy friends.

High neuroticism scores made it more likely to react to a situation with fear, anger, sadness, shame and anxiety; and such individuals often use maladaptive coping strategies (such as mood-altering drug use) [25] in stressful situations. Individuals with higher scores of neuroticism are more likely to lack emotional control and express themselves differently than adolescents with emotional stability on social networks. They are more likely to express ideas, emotions and problematic behaviors (such as substance use) on social networks openly and with limited concern for what others think of them [28,29]. Cyberbullies could use this information to harass them. Alcohol and tobacco use are a way to manage negative emotions among young people [30], which could expose them to anti-normative environments that foment getting in contact with cyberbullies.

Adolescents with low conscientiousness scores are more likely to be careless, discouraged, disorganized, depressed; with a worse global adult adjustment and mood-altering drug use [25]. In our study, conscientiousness was inversely associated with cybervictimization. Adolescents with low levels of conscientiousness are more likely to engage in risky behaviors and violent environments and to be less cautious in online behaviors (sharing passwords, using profiles on social networks with personal information). Hence, they are easier and more visible targets for cyberbullies.

Several limitations of this study are worth noting. First, data collected were obtained from self-reported information, although the confidentiality of the data was emphasized in order to obtain reliable measures. Secondly, the current analysis was limited to cross-sectional data. Although personality traits are relatively stable across an individual’s life, longitudinal design is needed to better address the causal relationship between personality traits, substance use and cybervictimization. Finally, we identified a cybervictim as any individual who suffered harassment “sometimes” during the previous year. This cut-off point may have been overly sensitive because it identified individuals who only suffered from infrequent harassment as cybervictims. However, we should also consider that traditional harassment and cyber harassment could have different consequences. Using shared websites for harassment can cause the victim to feel more threatened because it can be observed by more partners and last longer on the web. Nevertheless, we obtain significant results and the inclusion of these low levels of cyberbullying could bias the results toward the null hypothesis.

We also cannot distinguish between cybervictims and cyberbully-victims. Cyberbully-victims are youths who are both victims and perpetrators of cyberbullying. This group appears to show the most negative psychological and physical problems [31]. Despite these limitations, our study increases the limited amount of literature that addresses the relationship between substance use, cybervictimization and personality traits and represents an initial step towards understanding possible mechanisms underlying cybervictimization and substance use. Moreover, alcohol and tobacco use may be a coping strategy for cybervictimization used by adolescents, as in the case of traditional bullying [32].

The results of our study could help to design preventive actions against cyberbullying that are more effective and efficient through a better understanding of the victim’s personality traits. Preventive school programs could reduce rates of cybervictimization [33]. However, even with these promising results, this evidence was based on a few studies with certain methodological limitations [34].

Prevention programs for smoking or alcohol consumption in schools have had limited success [35,36], even if they were complex and involved families, teachers and schools [37]. A growing body of evidence suggests that generic (such as life skills, social skills and behaviour norms training) rather than substance-specific programmes are more effective [36]. Previous research has also found that interventions about coping skills targeting specific personality profiles reduced alcohol drinking [38] and illegal substance use [39].

## 5. Conclusions

Our study contributes to a better understand of cybervictimization and adolescent substance use through an understanding of the personality traits of adolescents and the associated risks of certain personality profiles. We found underlying common personality factors for cybervictimization and alcohol and tobacco use; thus, preventive programs aimed to identify students with these personality profiles and training them in coping skills could be effective and efficient to prevent cybervictimization and substance use.

## Figures and Tables

**Table 1 ijerph-16-03123-t001:** Characteristics of students who were and were not victims of cyberbullying.

Variables	Total Sample	Victims	Non-Victims	*p*-Value ^a^
	*n* (%)/Mean (SD)	*n* (%)/Mean (SD)	*n* (%)/Mean (SD)	
	765 (100%)	305 (39.87%)	460 (60.13%)	
Age	14.987 (0.66)	14.952 (0.67)	15.029 (0.65)	0.110
Sex				0.040
Female	432 (56.5%)	186 (43.1%)	246 (56.9%)	
Male	333 (43.5%)	119 (35.7%)	214 (64.3%)	
Mother’s education				0.874
Less than primary	22 (2.9%)	10 (3.4%)	12 (2.7%)	
Only Primary	197 (26.3%)	79 (26.5%)	118 (26.1%)	
Secondary	363 (48.4%)	146 (49.0%)	217 (48.0%)	
University	168 (22.4%)	63 (21.1%)	105 (23.2%)	
Father’s education				0.538
Less than primary	32 (4.3%)	12 (4.1%)	20 (4.5%)	
Only Primary	246 (33.3%)	105 (35.7%)	141 (31.7%)	
Secondary	362 (49.0%)	143 (48.6%)	219 (49.2%)	
University	99 (13.4%)	34 (11.6%)	65 (14.6%)	
Alcohol consumption				<0.001
Never or rarely	377 (49.5%)	117 (38.6%)	260 (56.8%)	
At least monthly	384 (50.5%)	186 (61.4%)	198 (43.2%)	
Tobacco consumption				0.008
Never	720 (94.7%)	281 (92.1%)	439 (96.5%)	
At least monthly	40 (5.3%)	24 (7.9%)	16 (3.5%)	
Personality traits (z-score)				
Openness	0 (1)	0.080 (0.985)	0.038 (0.952)	0.561
Conscientiousness	0 (1)	−0.001 (1.004)	0.200 (0.962)	0.007
Extraversion	0 (1)	0.110 (1.030)	−0.094 (0.948)	0.006
Agreeableness	0 (1)	0.049 (1.010)	0.246 (0.929)	0.73
Neuroticism	0 (1)	0.162 (1.017)	−0.234 (0.902)	0

^a^ Student *t*-test or Chi-square test.

**Table 2 ijerph-16-03123-t002:** Logistic regression models containing personality traits and drug consumption, predicting cybervictimization.

Variables	Model 1	Model 2	Model 3	Model 4	*p*-Value for Interactions ^a^
	OR (95% CI)	OR (95% CI)	OR (95% CI)	OR (95% CI)	
Age	1.305 (1.010–1.687)	1.200 (0.948–1.519)	1.195 (0.946–1.510)	1.263 (0.974–1.638)	0.743
Female	1.268 (0.882–1.822)	1.330 (0.978–1.809)	1.386 (1.023–1.829)	1.237(0.859–1.783)	-
Parental education					
Mother education (secondary or more)	1.060 (0.695–1.617)	1.008 (0.698–1.475)	0.999 (0.685–1.458)	1.056 (0.688–1.621)	0.962
Father education					
(secondary or more)	1.023 (0693–1.509)	1.136 (0.796–1.620)	1.141 (0.802–1.623)	1.035 (0.697–1.538)	0.347
Personality traits (z-score)					
Openness	1.070 (0.901–1.271)	-	-	1.079 (0.907–1.238)	0.274
Conscientiousness	0.730 (0.599–0.888)	-	-	0.779 (0.635–0.955)	0.611
Extraversion	1.454 (1.190–1.778)	-	-	1.314 (1.062–1.629)	0.458
Agreeableness	1.035 (0.835–1.284)	-	-	1.047 (0.843–1.301)	0.989
Neuroticism	1.575 (1.318–1.881)	-	-	1.526 (1.274–1.828)	0.484
Alcohol consumption					
At least monthly	-	1.993 (1.470–2.701)	-	1.506(1.053–2.154)	0.267
Tobacco consumption					
At least monthly	-	-	2.546 (1.284–5.048)	1.511 (0.727–3.143)	0.578

^a^*p*-Value for sex interactions terms in the fully adjusted logistic regression model (Model 4).

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
