# Peer review of "Alcohol and Tobacco Consumption, Personality, and Cybervictimization among Adolescents"

_ijerph, 2019, doi:10.3390/ijerph16173123_

Round 1

Reviewer 1 Report

Dear Authors

I find your contribution interesting and valid for cyberbullying research. Still, I have some serious concerns concerning your text:

Presentation of research overview is not sufficient, You present individual studies results concerning eg. gender and cybervictimization  as the results were consistent (eg. higher scores in girls). However results of many other studies show different results, also in meta-analyses. If you present so scarce data you should focus at least on the last kind of research. The same refers to the results on personality and victimization. If you use Problem Behaviour Theory, from my perspective, you should remember that problem behaviours tend to occur in clasters not necessarily influencing each other as independent/dependent variables. I think that the latter interpretation is too emphasized in the article in the unjustified way. Why there is so bi disproportion between boys and girls in the sample? What was the response rate? Has any official committee assessed the ethical dimension of the project. I have serious doubts on the smoking and alcohol consumption instruments. Categories there are not logically designed and overlapping. There are two categories in one question (frequency, and working days/weekends). Still it can not be changed at this stage. Almost absent is the interpretation that smoking/alcohol can be used as coping strategies for stress associated with cybervictimization. My serious concern (you also mention it in interpretation part) is inclusion as victims of anyone who indicated sometimes in one of the 12 categories during a year. It is contrary to bullying concept (repetition as a key characteristics). It is even more problematic due to the fact that severity of items varies significantly. I also suggest that talking about personality traits as simple independent factors, particularly in introductory part, is put in overdimplified way.

Author Response

Response to Reviewer 1 Comments

Point 1: I find your contribution interesting and valid for cyberbullying research. Still, I have some serious concerns concerning your text:

Presentation of research overview is not sufficient, you present individual studies results concerning eg. gender and cybervictimization as the results were consistent (eg. higher scores in girls). However, results of many other studies show different results, also in meta-analyses. If you present so scarce data, you should focus at least on the last kind of research.

Response 1: Thank you ever so much for your comments. Following your suggestions, we have provided a more detailed overview of the role of gender and cyberbulling. we have explained further in the introduction and we also agree to cited and highlight some meta-analyses: “Some studies found sex differences in cyberbulling and cybervictimization, girls were more likely than boys to experienced cyberbullying (Kowalski, R.M.; Limber, S.P. Electronic Bullying Among Middle School Students. J. Adolesc. Heal. 2007, 41, S22–S30; doi: 10.1016/j.jadohealth.2007.08.017). However, the existing evidence showed mixed results and the majority of studies did not found any sex differences(Gustafsson, E. Gender differences in cyberbullying victimization among adolescents in Europe. A systematic review. Malmö University: Faculty of health and society, Department of criminology, 2017. Available online: https://pdfs.semanticscholar.org/957d/ebcebb2b7ce867b11152aaa3cfec759a443a.pdf (accessed on 12 August 2019)”.

Point 2: The same refers to the results on personality and victimization. If you use Problem Behaviour Theory, from my perspective, you should remember that problem behaviours tend to occur in clusters not necessarily influencing each other as independent/dependent variables. I think that the latter interpretation is too emphasized in the article in the unjustified way.     

Response 2: Many thanks for this clarification, we agree that “Problem Behaviour Theory“ need to be explained. We have explained further in the introduction section: “These findings are consistent with the Problem Behaviour Theory (Jessor, R. Risk behavior in adolescence: A psychosocial framework for understanding and action. J. Adolesc. Heal. 1991, 12, 597–605; doi: 10.1016/1054-139X(91)90007-K). The conceptual structure of this theoretical framework includesdifferent domains (social environment, perceived environment, personality, behavior and biology) that are intercorrelate and determine clusters of risk behaviors that could be explained by similar antecedents (Looze 2014, aquesta es nova: de Looze, M.; ter Bogt, T.F.M.; Raaijmakers, Q.A.W.; Pickett, W.; Kuntsche, E.; Vollebergh, W.A.M. Cross-national evidence for the clustering and psychosocial correlates of adolescent risk behaviours in 27 countries. Eur. J. Public Health 2015, 25, 50–56; doi: 10.1093/eurpub/cku083). In line with this theory, adolescents who engage in antinormative behaviors, such as substance use, could easily break other rules (e.g. skipping class, getting into fights, shoplifting) and spend more time with antisocial peers with higher risk of getting involved in other problems as cybervictimization.”

And in the discussion section:“In consonance with Problem Behavior Theory (Jessor, R. Risk behavior in adolescence: A psychosocial framework for understanding and action. J. Adolesc. Heal. 1991, 12, 597–605; doi: 10.1016/1054-139X(91)90007-K), that points out how adolescent risk behaviors do not occur randomly but tend to cluster and share common psychosocial risk factors, adolescents with higher levels of neuroticism or extraversion or low level of conscientiousness could use coping strategies that increases risky behaviors associated both with substance use and cybervictimization. In this sense, personality could be a common cause for being cybervictimized and for consumption of alcohol and tobacco. Moreover, after removing the effect of personality by multivariate analysis, we observed alcohol consumption remained also independently associated of being victimized”.

Point 3: Why there is so disproportion between boys and girls in the sample?

Response 3: In the baseline data there were boys as girls were matched (49.4% vs. 50.6%). However, at 4 years follow-up study, the number of girls was greater than men (56.5% vs 43.5%). The ITACA study had a baseline assessment 2011-12 and 3 waves in 2013; 2014; 2015. We believed there were sex differences in the response rate, because girls miss less days of school and respond on more accurate way the questionnaire that better allow to identify then. In fact, most of the questionnaires discarded as incomplete data were full filled by boys.

This sex distribution is consistent with different studies (60% girls vs 40% boys), including meta-analysis (Fisher, BW, Gardella, JH, & Teurbe-Tolon, AR (2016). Peer Cybervictimization Among Adolescents and the Associated Internalizing and Externalizing Problems: A Meta-Analysis. Journal of Youth and Adolescence, 45 (9), 1727-1743. Doi: 10.1007 / s10964-016-0541-z).  

Point 4: What was the response rate?

Response 4: The sample are from the ITACA study a multi-centre, cluster-randomized controlled trial, aimed at reducing the prevalence of smoking among secondary education students, there were 3 waves (cross-sectional surveys at schools) during the study, in the third wave participate 1230 students and 406 were not matched with the baseline data and not included and 59 were not valid data because the questionnaire was incomplete, the response rate was 62.2%.

We agree with the reviewer and we include this information and we have explained further in the sample and procedure section: “The study participants were students aged 14–16 years, who participated in the project ITACA: a multi-centre, cluster-randomized controlled trial, aimed at reducing the prevalence of smoking among secondary education students (Leiva, A., Estela, A., Torrent, M., Calafat, A., Bennasar, M., Yanez, A., 2014. Effectiveness of a complex intervention in reducing the prevalence of smoking among adolescents: study design of a cluster-randomized controlled trial. BMC Public Health 14:373). The initial ITACA sample comprised 1708 students (11-12 year-old) of 16 secondary education schools covering a wide range of communities (urban, semi urban and rural), socioeconomic status and prevalence of smoking. Schools were randomly assigned to a 4-year curriculum based multifactorial intervention or control groups. Here we focused on the third wave of assessment (September-December 2015) when personality and cybervictimization was assessed (1230 students). Participants met the inclusion criteria if they attended school on the day of the survey, their parents agreed with participation in the study and student could be identified and matched with baseline data. The final sample comprised of 765 students“.

Point 5: Has any official committee assessed the ethical dimension of the project?

Response 5: The study protocol was approved by the Balearic Islands Health Research Ethics Committee (CEI-IB, registration number: 1146/09PI).

 We have explained further in the manuscript: “The research protocol was approved by the Primary Care Research Committee and the Institutional Review Board of the Balearic Islands Health Service (CEI-IB Ref. No: 1146/09PI). The study was conducted according to the ethical guidelines of the Declaration of Helsinki. Written informed consent was obtained from all students and at least one parent per student. And, all materials and procedures were approved by the educational authority”.

Point 6: I have serious doubts on the smoking and alcohol consumption instruments. Categories there are not logically designed and overlapping. There are two categories in one question (frequency and working days/weekends). Still it cannot be changed at this stage.

Response 6: We agree with the reviewer that some categories overlap, however adolescents were asked to mark the statement (only one) that best describes them, Because the ITACA study is randomized clinical trial aimed to prevent smoking in adolescents, the main outcome of the study should be measured by a validate instrument, there are not many validate questionnaire about tobacco consumption for adolescents In Spain. We chose the ESFA questionnaire (Longitudinal effects of the European smoking prevention framework approach. ESFA project in Spanish adolescents. Ariza C, Nebot M, Tomás Z, Giménez E, Valmayor S, Tarilonte V, De Vries H. Eur J Public Health. 2008 Oct;18(5):491-7) previously validated in Spanish population (Comin BE, Torrubia BR, Mor SJ, Villalbi H Jr, Nebot AM: The reliability of a self-administered questionnaire for investigation of the level of exercise, smoking habit and alcohol intake in school children. Med Clin (Barc.) 1997, 108:293–298). 

We did not find any short-validated questionnaire for alcohol consumption for adolescents, we decided to adapt the Spanish version of the questionnaire for alcohol consumption from the European social surveys:

(https://www.europeansocialsurvey.org/docs/round7/fieldwork/source/ESS7_source_main_questionnaire.pdf)

Spanish version: (https://www.europeansocialsurvey.org/docs/round7/fieldwork/spain/spanish/ESS7_questionnaires_ES_spa.pdf).

Point 7: Almost absent is the interpretation that smoking/alcohol can be used as coping strategies for stress associated with cybervictimization.

Response 7: Following your suggestion we have added the interpretation of coping strategies for stress in the discussion section: “Moreover, alcohol and tobacco may be used by adolescents as coping strategy for the cybervictimization, as in the case of traditional bullying”.

Point 8: My serious concern (you also mention it in interpretation part) is inclusion as victims of anyone who indicated sometimes in one of the 12 categories during a year. It is contrary to bullying concept (repetition as a key characteristics). It is even more problematic due to the fact that severity of items varies significantly.

Response 8: We also agree with the reviewer and we added a justification in the discussion section: “However, we should also consider that traditional harassment and cyber harassment could have different consequences. Using shared websites for harassment can cause the victim to feel more threatened because it can be observed by more partners and last longer on the web.  Nevertheless, the inclusion of these low levels of cyberbullying could bias the results toward the null hypothesis and we get significant results”.

Point 9: I also suggest that talking about personality traits as simple independent factors, particularly in introductory part, is put in oversimplified way. 

Response 9: We agree with the reviewer that there was some incoherence between the theorical framework and the interpretation of our results. Please find above (second answer) the mentioned reviewed section.

Reviewer 2 Report

-They do not justify the reason why the age of the participants was selected. 2015 data and a project that does not provide information.

- does not indicate how the study centers were selected, does not indicate the areas of Spain. Instead, it provides a reference for the year 2014. Do you mean that it is a part of that study?

 -all seem to indicate that it was a study with other objectives.

- It does not detail where the questions about drug use have been extracted.

- does not justify questions about drug use taken from a test. does not provide validity data.

 - So the objective of the study is not justified. which even means that no differences were found between alcohol consumption and personality.

Author Response

Response to Reviewer 2 Comments

Point 1: They do not justify the reason why the age of the participants was selected. 2015 data and a project that does not provide information.

Response 1: The source population were students attending the first course of compulsory secondary education (11 to 13 years-old) that were included in the ITACA study (Leiva, A., Estela, A., Torrent, M., Calafat, A., Bennasar, M., Yanez, A., 2014. Effectiveness of a complex intervention in reducing the prevalence of smoking among adolescents: study design of a cluster-randomized controlled trial. BMC Public Health 14:373). ITACA study is a multi-centre, cluster-randomized controlled trial, aimed at reducing the prevalence of smoking among secondary education students (14-16 years-old). We have expanded this information in the sample and procedure section: “The study participants were students aged 14–16 years years, who participated in the project ITACA: a multi-centre, cluster-randomized controlled trial, aimed at reducing the prevalence of smoking among secondary education students[18]. The initial ITACA sample comprised 1708 students (11-12 year-old) of 16 secondary education schools covering a wide range of communities (urban, semi urban and rural), socioeconomic status and prevalence of smoking. Schools were randomly assigned to a 4-year curriculum based multifactorial intervention or control groups. Here we focused on the third wave of assessment (September-December 2015) when personality and cybervictimization was assessed (1230 students). Participants met the inclusion criteria if they attended school on the day of the survey, their parents agreed with participation in the study and student could be identified and matched with baseline data. The final sample comprised of 765 students”.

Point 2: Does not indicate how the study centers were selected, does not indicate the areas of Spain. Instead, it provides a reference for the year 2014. Do you mean that it is a part of that study?

Response 2: We have explained further in the sample section:“The study participants were students aged 14 to 16 years, who participated in the project ITACA: a multi-centre, cluster-randomized controlled trial, aimed at reducing the prevalence of smoking among secondary education students”.

We randomly selected 22 study centers from 26 municipalities in the Balearic Islands. If the selected school declined to participate, another school from the same municipality was invited. There were 2,527 students who attended first course in these 22 participating schools and 2,404 of them (95.1%) agreed to participate. However, 6 schools dropped out following study arm assignment. We have expanded this information in the sample section.

Please find more detailed information in the published protocol design (Leiva, A., Estela, A., Torrent, M., Calafat, A., Bennasar, M., Yanez, A., 2014. Effectiveness of a complex intervention in reducing the prevalence of smoking among adolescents: study design of a cluster-randomized controlled trial. BMC Public Health 14:373).

Point 3: All seem to indicate that it was a study with other objectives.

Response 3: The sample of our study were student who participated in ITACA study. However, there were some secondary objectives different from the evaluation of the intervention, for the present manuscript a research protocol was designed an approved by the principal investigator to test the hypothesis that cybervictimization was associated with personality, tobacco and alcohol use and could be potential mediated by sex.

Point 4: It does not detail where the questions about drug use have been extracted.does not justify questions about drug use taken from a test. does not provide validity data.

Response 4: The ITACA study is randomized clinical trial aimed to prevent smoking in adolescents, the main outcome of the study should be measured by a validate instrument.
Smoking was measured by a previously validated questionnaire for adolescents (Comin BE, Torrubia BR, Mor SJ, Villalbi H Jr, Nebot AM: The reliability of a self-administered questionnaire for investigation of the level of exercise, smoking habit and alcohol intake in school children. Med Clin (Barc.) 1997, 108:293–298). Information on tobacco use was be collected through the following question: Which of the following statements best describes you? (A) I have never tried to smoke; (B) I have tried cigarettes a few times, but I do not smoke now; (C) I currently smoke less than one cigarette per month; (D) I currently smoke at least one cigarette per month, but less than one cigarette per week; (E) I currently smoke at least one cigarette per week; (F) I smoke every day; (G) I used to smoke regularly in the past, but I do not smoke now.

We did not find any short-validated questionnaire for alcohol consumption for adolescents, we decided to adapt the Spanish version of the questionnaire for alcohol consumption from the European social surveys:

(https://www.europeansocialsurvey.org/docs/round7/fieldwork/source/ESS7_source_main_questionnaire.pdf).

Spanish version:

(https://www.europeansocialsurvey.org/docs/round7/fieldwork/spain/spanish/ESS7_questionnaires_ES_spa.pdf).

Point 5: So, the objective of the study is not justified. which even means that no differences were found between alcohol consumption and personality.

Response 5: The objective of the study was to examine the association between cybervictimization with personality, tobacco and alcohol use and evaluate potential sex mediated differences. We found that personality, tobacco and alcohol use were associated with cybervictimization. However, smoking was not independent associated with cybervictimization in the multivariant analysis and we conclude that personality could be a confounder for this association. However, alcohol and personality were independently associated with cybervictimization and could be directly related with cybervictimization. Furthermore, we did not find sex differences in the association between studied risk factors and cybervictimization.

Round 2

Reviewer 1 Report

Dear Authors

Thank you for accepting my suggestions and the answers.

Good luck

Reviewer 2 Report

- the authors have made the suggested changes.

- the manuscript has improved substantially